# A Rare and Easily Overlooked Case of Bilateral Traumatic Testicular Dislocation and an Alternative Viewpoint on Delayed Management

**DOI:** 10.3390/medicina59050892

**Published:** 2023-05-06

**Authors:** Ming-Wei Hsu, Po-Fan Hsieh, Li-Hsien Tsai

**Affiliations:** 1Department of Urology, China Medical University and Hospital, Taichung 404, Taiwan; d29950@mail.cmuh.org.tw (M.-W.H.); d17341@mail.cmuh.org.tw (P.-F.H.); 2School of Medicine, China Medical University, Taichung 404, Taiwan

**Keywords:** traumatic testicular dislocation, testicular outcomes, surgery

## Abstract

The incidence of traumatic testicular dislocation is rare, and it is usually overlooked in an initial diagnosis. We present a case of bilateral dislocated testes after a traffic accident that was treated via orchidopexy one week later. No testicular complications had occurred by the time of the follow-up visit. Generally, surgery is often postponed owing to a late diagnosis or another major organ injury, and the adequate timing of surgery is still under debate. We performed a review of past cases, which showed similar testicular outcomes irrespective of surgical timing. Delayed intervention may be a feasible decision after a patient’s hemodynamic status is stable for surgery. To prevent delayed diagnosis, scrotal examination should not be overlooked in any patients presenting with pelvic trauma to the emergency department.

## 1. Introduction

Traumatic testicular dislocation refers to the displacement of a testis following scrotal blunt trauma. It is an uncommon complication after pelvic trauma, especially with respect to motorcycle collisions or straddle injuries. The first recorded case dates back to an 1809 report by Claubry [1]. To date, there have been fewer than 200 reported cases worldwide, and many cases may remain unreported due to the difficulty in diagnosis.

The diagnosis is usually missed initially due to the injury being masked by other major trauma such as pelvic fractures or intra-abdominal bleeding, and the diagnosis relies on scrotal examinations and imaging modalities [2]. Moreover, it can be challenging to distinguish traumatic dislocated testes from undescended testes and retractile testes due to their similar presentations. As a result, a patient’s medical history is also crucial for a definite diagnosis [1].

How to improve the detection rate for this injury is an important question that remains unsolved. Early repositioning can only be performed if an early diagnosis is made, and delayed management may result in complications such as atrophy, infertility, and even malignant changes based on prior cases [3,4,5]. Whether delayed reduction affects testicular outcomes is still controversial. We present a case with traumatic testicular dislocation who received a delayed surgical intervention. We will provide our perspective on preventing delayed diagnosis and review the testicular outcomes in previous cases. To the best of our knowledge, this is the first article to outline past outcomes after treatment, including atrophy, recurrence, infertility, and severe complications.

## 2. Case Report

The examined 33-year-old male had no medical history and presented to the emergency department with pelvic pain after a motor vehicle collision. A physical examination revealed a normal Glasgow Coma Scale score and multiple abrasion wounds across the trunk, all four limbs, and scrotum. Computed tomography (CT) showed traumatic type B aortic dissection; fractures across the pelvic bone (open book pelvic fracture), left acetabulum, and sacrum (left sacroiliac joint diastasis); and bilateral distal radius fractures (Figure 1). A urologist was initially consulted due to difficulties with Foley catheter insertion. Under a stable hemodynamic status, the patient received emergent percutaneous thoracic endovascular aortic repair (pTEVAR) surgery for traumatic type B aortic dissection and was admitted to the intensive care unit. One day after surgery, bilateral bulging inguinal masses were noted, and we also failed to identify bilateral testes. The patient stated that both testes were in the scrotum before the accident. The abdominal CT scan was reviewed again, and bilateral inguinal testes were found with surrounding air formation (Figure 2). The urologist was consulted again, and sonography showed bilateral intact testes with normal blood flow in inguinal areas. However, manual reduction of both testes was unsuccessful. One week after admission, the patient was treated using the Stoppa operation for pubic symphysis and left sacroiliac (SI) joint fractures. Bilateral orchiopexy was performed at the same time. During the operation, we made a transverse suprapubic incision and freed the testes from the inguinal canal. Both testes over the external ring were grossly healthy (Figure 3). We then opened the scrotum through the midline and pulled the testes to the scrotum with fixation. The patient was discharged on day 20 after admission without further traumatic or surgical complications. Six months after the surgery, scrotum sonography showed the correct position of the bilateral testes with preserved blood flow. A scrotal examination showed normal size and position of both testes without signs of atrophy (Figure 4). Written informed consent was provided by the patient for publication of the case.

## 3. Discussion

During a motor vehicle collision, the sudden deceleration causes the scrotum to collide with the fuel tank, generating blunt impact that displaces the testes, while the simultaneous contraction of cremasteric muscles amplifies the applied force. Pressure exerted on the scrotum during trauma can push the testes back into the inguinal canal (in around 50% of cases), pubic and penile areas, and even into the abdomen [6]. Bilateral dislocation accounts for approximately one third of all cases [7]. The possible predisposing factors include small testes, a history of a previous inguinal hernia, and an open external ring [1].

Genitourinary injuries account for about 10 percent of cases of abdominal trauma [8]. While trauma patients may present with multi-organ injuries that include the scrotum, physicians typically begin their assessment by evaluating vital organs. Urethral injuries may be identified early due to the urgent need for a Foley catheter to measure urine output, and urologists would be consulted if there is blood at the meatus or difficulty inserting a urethral catheter. In their rush to diagnose and treat major trauma, physicians may overlook a scrotal examination if there are no obvious abnormalities in the external genitalia. At times, cardiopulmonary resuscitation is even needed in hemodynamically unstable patients, and dislocated testes are often found either by medical personnel or the patient after initial management. At the first presentation, an inguinal mass with an empty scrotum may be noted. However, the condition may be asymptomatic, and cases have been reported up to 15 years after the initial injury [9]. Diagnostic tools include ultrasound and abdominal CT. Ultrasound can be used to perform the preliminary diagnosis of ectopic testes and confirm the testes’ integrity and current blood flow. Abdominal CT is the most precise modality available, and it can provide details of the location of the testes, associated injuries such as ruptures, and the relationship with adjacent anatomy. However, initially, the testes are often neglected due to the non-fatal nature of the corresponding injury and a lack of awareness of the dislocation, even when CT scans reveal dislocated testes in patients with pelvic trauma. In our case, a urologist was initially consulted for Foley catheter insertion; however, the condition in question was not identified, and orchidopexy was subsequently postponed. As a result, we strongly recommend performing a scrotal examination for every patient who has experienced pelvic trauma, especially a straddle collision. This simple and fast examination takes only a few seconds and helps to quickly identify the dislocation of the testes.

The treatment options for this injury include manual reduction and surgical fixation. Manual reduction has a relatively low success rate due to tissue swelling, small fascia defects, and patient intolerance [3]. It may even cause further damage to the testes, including torsion. Surgical fixation provides a definite solution, during which the testes can be exposed, allowing for a thorough examination followed by repositioning [7]. Immediate surgical exploration is necessary in patients with severe testicular complications, such as rupture and torsion. The current recommendations for testicular rupture involve surgical exploration within 3 days, which can save testicular function and increase salvage rates up to 90 percent [10]. Testicular torsion is defined as a twisted spermatic cord that reduces blood supply to the testes. Surgical exploration within 6 h of onset can save up to 90 percent of testes, but postponing surgery can decrease that to 10 percent [11]. Scrotal ultrasound is readily available and is the modality of choice for identifying testicular rupture and torsion. In cases of traumatic dislocated testes, most patients receive early surgery if a prompt diagnosis is made. However, delayed surgery may occur due to the following reasons: (1) a late diagnosis was made during the secondary survey; (2) the patient was initially reluctant to receive surgery; (3) symptoms arose weeks or even months after the accident; and/or (4) the patient was hemodynamically unstable. Delayed surgery may correlate with potential complications, including testicular discomfort, torsion, ischemia, atrophy, impaired spermatogenesis, and even infertility [12]. Nevertheless, atrophy and hypospermatogenesis are reported to be reversible after delayed management in some cases [9,13]. Consequently, whether surgical timing affects testicular outcome is controversial, as there is a lack of comparative studies available from which to draw definitive conclusions [14].

We performed a non-systematic search of Medline and Google Scholar using the key words testis dislocation or testicular dislocation and excluded articles without full context and those not in English. Sixty-three articles were identified, consisting of eighty-nine cases from 1951 to 2022. The distribution of laterality for dislocated testes was bilateral in 27 cases (30.6%), right-sided in 38 cases (42.7%), and left-sided in 24 cases (26.7%). The majority of dislocated testes were located above the groin (58/89), while others were found in the pubic area, abdomen, pelvis, perineum, thigh, and even hip joint. Of the 89 patients, 73 patients underwent surgery, of whom 30 had surgery within one week and 35 had delayed surgery owing to a misdiagnosis or unstable condition, and the timing of surgery was not mentioned for the remaining 8 patients. The details of the previously reported cases are presented in Table 1.

The timing of surgery was variable in the reviewed cases, and whether early reduction of the dislocated testes made any difference compared to delayed reduction remained inconclusive. Among the reviewed articles, 41 of the 73 patients had follow-up data. Surprisingly, neither atrophy nor recurrent dislocation were recorded in all patients after surgery. Atrophic testes were sometimes noted at first, but had then recovered after intervention. [9] Our patient underwent testicular reduction one week after the diagnosis, and both testes remained intact during a follow-up. Considering our case and the reviewed cases, delayed reduction does not seem to affect the eventual testicular size or recurrence rate. However, operative repair that is delayed for a period of weeks may be rendered increasingly difficult by scarring and fibrosis around the testes.

On the other hand, many patients were concerned about infertility after long-term dislocation. A few of the reviewed cases underwent a biopsy of the testes, which showed variable results (Appendix A). Some patients experienced hypospermatogenesis upon diagnosis, but seminal analysis improved after surgical reduction. Two patients even suffered from infertility after their respective accidents. However, both had spontaneous pregnancies with their partner after receiving treatment [9,13]. Sakamoto et al. reported a case involving a patient who had bilateral dislocated testes for 15 years [9]. The patient presented with infertility and abnormal spermatogenesis initially, but then recovered after reduction; subsequently, his wife had a spontaneous pregnancy. Hayami et al. also reported a case of a patient who recovered spermatogenesis after a 16-week delay in testis reduction after dislocation [14]. Taken together, our findings show similar seminal outcomes regardless of surgical timing and the restoration of spermatogenesis, even after prolonged testicular dislocation. It would be valuable to further investigate the relationship between testicular dislocation and seminal function in future studies.

With regard to associated complications, 14 (15.7%) patients suffered severe injuries after their respective accidents. Four of them had a testicular rupture, eight patients suffered from testicular torsion, one patient had partial testicular necrosis one year after testicular dislocation, and the last patient suffered from an extensive injury of the right testis after compression due to a large log. Additionally, five (5.6%) patients underwent an orchiectomy due to irreversible testicular injury or rupture. Scrotal trauma is divided into penetrating injuries and blunt injuries. Testicular dislocation is an uncommon presentation; however, testicular rupture occurred in 50 percent of patients with blunt scrotal trauma [15]. In our review, only four patients with testicular rupture were present in all the cases of traumatic dislocated testes, which can be explained by the protective mechanism of the scrotum and testes, including with respect to the fact that (1) testes can move freely within the elastic scrotum and slip away from direct external compression, (2) the forces from the collision and cremasteric reflex retract the testes through the wide external ring, and (3) the durable and fibrous layer of the tunica albuginea protects testes from injury [15]. If a testis was pushed towards the external ring during a collision, it was often squeezed into the inguinal canal and other uncommon locations with a low incidence of severe complications.

## 4. Conclusions

Dislocated testes may not be identified at first. A scrotal physical exam is strongly recommended for pelvic trauma patients in order to exclude testicular injury. The timing of operation may not correlate with testicular outcomes, including atrophy, the recurrence rate, and seminal results, but further research is needed. Nevertheless, emergency surgery is warranted if a severe testicular injury exists.

## Figures and Tables

**Figure 1 medicina-59-00892-f001:**
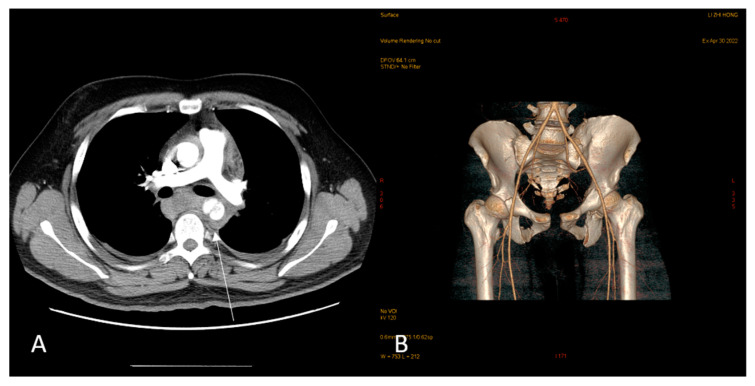
Images captured at emergency department: Aortic dissection (**A**) and pelvic bone fracture (**B**).

**Figure 2 medicina-59-00892-f002:**
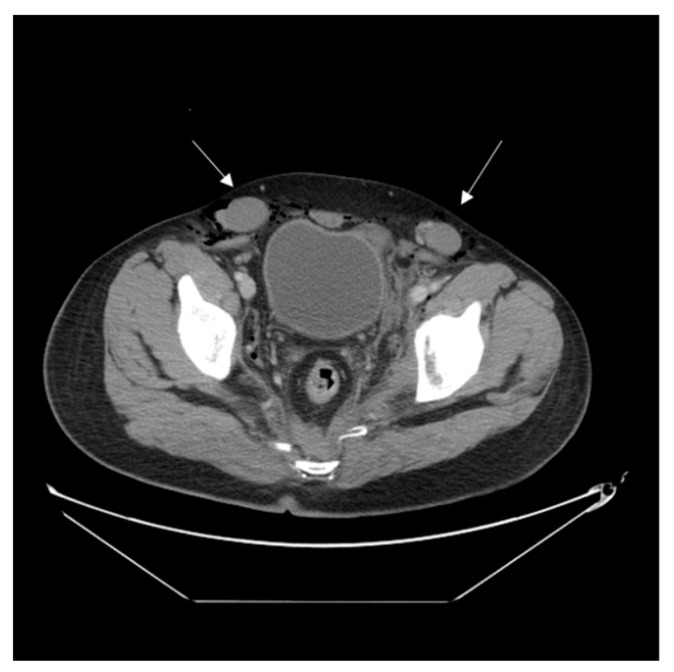
Abdominal CT scan: The arrows indicated bilateral dislocated testes.

**Figure 3 medicina-59-00892-f003:**
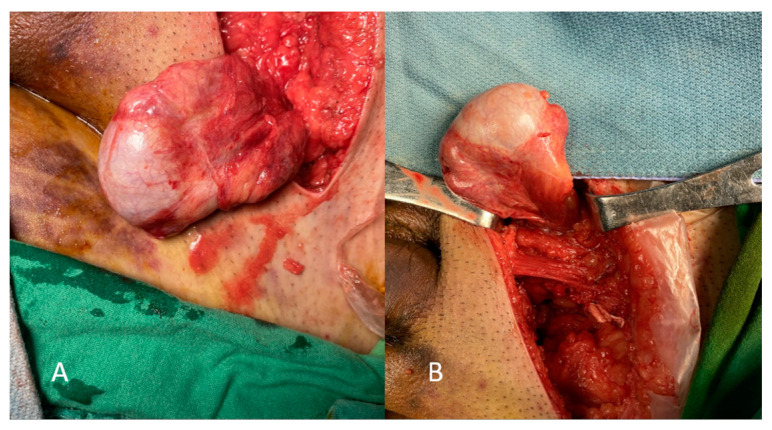
Left testis (**A**) and right testis (**B**) during surgery.

**Figure 4 medicina-59-00892-f004:**
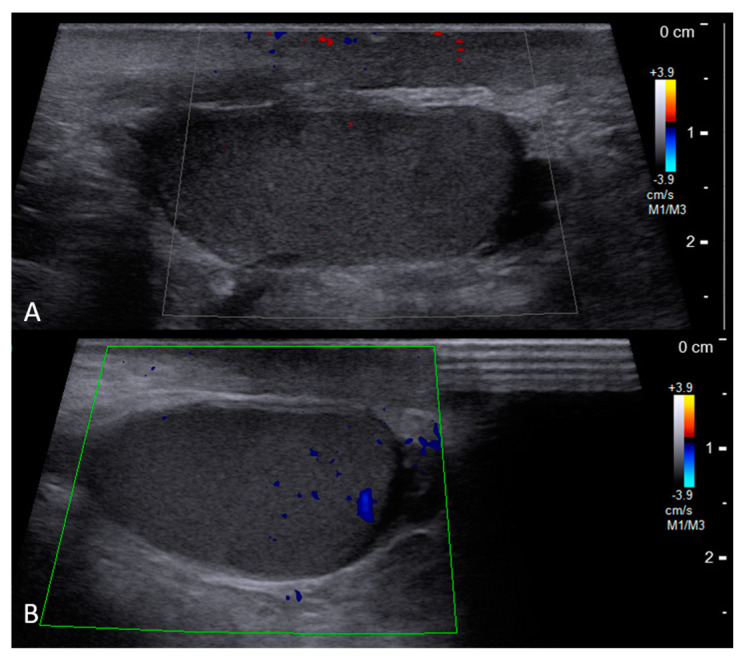
Follow−up scrotal sonography: left testis (**A**) and right testis (**B**).

**Table 1 medicina-59-00892-t001:** Reported cases with testicular dislocation.

No.	Author/Year	Cases	Laterality	Location	Management	Testicular Complications
1	Charnock, 1951	1	Right	Groin	Delayed orchidopexy	No
2	Morgan, 1965	4	Right: 3 Left: 1	Groin	Early orchidopexy: 1 Delayed orchidopexy: 2 Manual reduction: 1	No
3	Neistadt, 1967	1	Bilateral	Pubic area	Manual reduction	No
4	Sethi, 1967	2	Left: 1 Right: 1	Prepuce: 1 Unknown: 1	Delayed orchidopexy: 1 Observation: 1	No
5	Boardman, 1975	1	Bilateral	Groin	Early orchidopexy	No
6	Goulding, 1976	2	Right	Groin	Delayed orchidopexy	No
7	Edson, 1979	1	Bilateral	Right groin and left pubic area	Early orchidopexy and testicular repair	Left rupture
8	Kauder, 1980	1	Bilateral	Groin	Manual reduction: left Delayed orchidopexy: right	No
9	Foster, 1981	1	Right	Groin	Orchidopexy with unreported timing	No
10	Pollen, 1982	1	Bilateral	Groin	Delayed orchidopexy and detorsion	Bilateral 180° rotation
11	Nakarajan, 1983	3	Right: 1 Left: 1 Bilateral: 1	Groin	Delayed orchidopexy	No
12	Koga, 1990	1	Bilateral	Groin	Early orchidopexy	No
13	Singer, 1990	1	Right	Groin	Manual reduction	No
14	J Ishikawa, 1990	1	Right	Groin	Delayed orchidopexy	No
15	Feder, 1991	1	Right	Intra-abdomen	Early orchidopexy	No
16	Lee, 1992	2	Right	Groin: 1 Pubic area: 1	Delayed orchidopexy	No
17	Wright, 1993	1	Right	Groin	Delayed orchidopexy	No
18	Madden, 1994	1	Right	Groin	Manual reduction	No
19	Schwartz, 1994	1	Right	Groin	Orchidopexy with unreported timing	Hematoma
20	Toranji, 1994	1	Right	Anterior abdominal wall	Orchidopexy with unreported timing	No
21	Hayami, 1996	1	Right	Groin	Delayed orchidopexy and detorsion	180° rotation
22	O’Donnell, 1998	3	Bilateral:1 Left: 1 Right: 1	Groin: 1 Contralateral scrotum: 1 Pubic area: 1	Orchidopexy with unreported timing	No
23	Yagi, 1999	1	Left	Thigh	Delayed orchidopexy and detorsion	180° rotation
24	Shefi, 1999	1	Left	Groin	Early orchidopexy	No
25	Yoshimura, 2002	1	Bilateral	Groin	Delayed orchidopexy and detorsion	Bilateral 180° rotation and atrophy
26	Bromberg, 2003	1	Bilateral	Groin	Manual reduction: right Delayed orchidopexy: left	No
27	Blake, 2003	1	Right	Groin	Orchidopexy with unreported timing	No
28	Chang, 2003	1	Right	Groin	Manual reduction	No
29	O’Brien, 2004	1	Bilateral	Groin: left Retro-vesical area: right	Early orchidopexy and testicular repair	Left rupture
30	Ko, 2004	9	Bilateral:2 Right: 4 Left: 3	Groin: 3 Pubic area:5 Penis:1	Delayed orchidopexy: 5 Orchiectomy: 1 Manual reduction: 3	One torsion and infarction
31	Bedir, 2005	1	Right	Perineum	Delayed orchidopexy	-
32	Sakamoto, 2008	1	Bilateral	Groin	Delayed orchidopexy and detorsion	Bilateral 180° rotation and atrophy
33	Ezra, 2009	1	Bilateral	Groin	Early orchidopexy	NO
34	Kilian, 2009	1	Bilateral	Groin	Early orchidopexy	No
35	Aslam, 2009	1	Left	Groin	Delayed orchidopexy	No
36	Vasudeva, 2010	1	Right	Groin	Early orchidopexy	No
37	Perera, 2011	1	Left	Groin	Early orchidopexy and detorsion	Partial torsion
38	Tsurukiri, 2011	1	Bilateral	Perineum	Orchidopexy with unreported timing	No
39	Naseer, 2012	1	Left	Groin	Early orchidopexy	No
40	Smith, 2012	1	Bilateral	Groin	Early orchidopexy	No
41	Sinasi, 2012	1	Left	Groin	Manual reduction	No
42	Boudissa, 2013	1	Bilateral	Intrapelvic: left Groin: right	Early orchidopexy	No
43	Matzek, 2013	1	Right	Groin	Early orchidopexy	No
44	Zavras, 2014	1	Left	Groin	Early orchidopexy	No
45	Gómez, 2014	7	Left: 4 Right: 1 Bilateral: 2	Groin: 5 Suprapubic: 4	Manual reduction:1 Early orchidopexy: 2 Delayed orchidopexy: 2 Mortality: 2	No
46	Pesch, 2014	1	Right	Groin	Early orchidopexy	Atrophy
47	Meena, 2014	1	Left	Groin	Manual reduction	No
48	Wiznia, 2016	1	Left	Groin	Early orchidopexy	No
49	Kim, 2016	1	Right	Groin	Orchiectomy	Partial necrosis
50	Carvalho, 2018	1	Bilateral	Groin	Delayed orchidopexy	No
51	Shirono, 2018	1	Bilateral	Groin	Manual reduction failed, with spontaneous descending 3 days later	No
52	Middleton, 2019	2	Bilateral:1 Right:1	Groin	Manual reduction: 1 Delayed orchidopexy: 1	No
53	Lenfant, 2019	1	Right	Pubic area	Orchiectomy	Rupture
54	Montes, 2020	1	Right	Groin	Early orchidopexy	No
55	Subramania, 2020	1	Bilateral	Groin	Early orchidopexy	Left epididymal hematoma
56	Lovsin, 2020	1	Bilateral	Externalized: left Hip joint: right	Right orchiectomy Left partial resection and early orchidopexy	Bilateral extensive injuries
57	Bernhard, 2021	1	Left	Pubic area	Early orchidopexy	No
58	Mangual-Perez, 2021	1	Left	Abdomen	Early orchidopexy	No
59	Naik, 2021	1	Bilateral	Root of the scrotum: left Groin: right	Early orchidopexy	No
60	Chouhan, 2021	1	Left	Intra-pelvic	Early orchidopexy	No
61	Wang, 2021	1	Right	Groin	Delayed orchidopexy	No
62	Chiu, 2022	1	Right	Groin	Orchiectomy	Rupture
63	Saeedi, 2022	2	Right: 1 Left: 1	Groin	Delayed orchidopexy: 1 Early orchidopexy and detorsion: 1	Torsion

## Data Availability

The data presented in this study are available on reasonable request from the corresponding author.

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
