# Peer review of "A Rare and Easily Overlooked Case of Bilateral Traumatic Testicular Dislocation and an Alternative Viewpoint on Delayed Management"

_medicina, 2023, doi:10.3390/medicina59050892_

Round 1

Reviewer 1 Report

Testicular dislocation following trauaa is extremely seldom found. The authors present 1 case. The case report should be better structured and address only dislocation and not other entities e.g. rupture etc.

Images should be related to topic of dislocation

Reviewer 2 Report

Authors should be congratulated for their work. The topic is rare and a higher appropriate medical awareness is needed. The manuscript is well written but several improvements are required. Why did not Authors perform a systematic review of current evidence? The non-systematic search performed should be properly described in a material and methods section. Moreover, a PRISMA is required, in my opinion, to describe the available evidence on the topic. Specifically, it is more easy to read first the non-systematic search and then the presentation of the case report. 

Related to the non-systematic search, are data available on the outcome comparison between early and delayed orchydopessi ??? They could add a valuable point of view to the current manuscript.  Are data available on fertility status after the accident? Did a spinal injury occur? (PMID: 35743658).

A major revision is required.

Round 2

Reviewer 1 Report

The comments were not completely addressed in the revised manuscript

Author Response

Dear reviewer,

We have made modifications to the content related to " testicular rupture and torsion" from line 116 to 125 in the revised manuscript. We acknowledge that ruptures and torsions may correlate with testicular dislocation, as they are complications from testicular trauma and they have ever co-existed in previous cases. Additionally, when rupture or torsion happens, immediately surgery is necessary. We deemed it necessary to emphasize the importance of these complications and enrich our case report.  We have removed any extraneous information and have made the article more concise. The adjustment are shown in the discussion section.

Regarding the images, figure 1 demonstrates the severity of the trauma that can occur when testicular dislocation happens. Testicular dislocation was masked initially by aortic dissection and multiple fractures in our patient. As a result, in patients with major trauma, testicular problem should not be ignored. We hope to address the issue that testicular dislocation can easily be masked by other major injuries.

Thanks for your recommendations.

Reviewer 2 Report

The authors modified properly the manuscript which now is suitable for publication. 

Author Response

Dear reviewer,

Thanks for your comments.